# Sorption of Polar Sorbents into GO Powders and Membranes

**DOI:** 10.3390/membranes13010053

**Published:** 2023-01-01

**Authors:** A. V. Kaplin, A. T. Rebrikova, E. A. Eremina, N. A. Chumakova, N. V. Avramenko, M. V. Korobov

**Affiliations:** 1Chemistry Department, M.V. Lomonosov Moscow State University, Leninskiye Gory, 1/3, Moscow 119991, Russia; 2N.N. Semenov Federal Research Center for Chemical Physics, Russian Academy of Science, Kosygin St. 4, Moscow 119991, Russia

**Keywords:** graphite oxide, powders, membranes, sorption, polar adsorbents

## Abstract

The comparative study of sorption of polar substances acetonitrile and water into powders and membranes (>10 μm thick) of modified Hummers (HGO) and Brodie (BGO) graphite oxides was performed using isopiestic method (IM) and differential scanning calorimetry (DSC). Additional sorption data were obtained for pyridine and 1-octanol. Sorption measurements were accompanied by conventional XRD and XPS control. Electron paramagnetic resonance (EPR) was additionally used to characterize ordering of the membranes. The impact on sorption of synthetic procedure (Brodie or Hummers), method of making membranes, chemical nature of the sorbent, and method of sorption was systematically examined. It was demonstrated that variations in synthetic procedures within both Hummers and Brodie methods did not lead to changes in the sorption properties of the corresponding powders. Sorption of acetonitrile and pyridine was reduced by approximately half when switching from powders to membranes at ambient temperature. DSC measurements at a lower temperature gave equal sorption of acetonitrile into HGO powder and membranes. Water has demonstrated unique sorption properties. Equal sorption of water was measured for HGO membranes and powders at T = 298 K and at T = 273 K. It was demonstrated that lowering the orientational alignment of the membranes led to the increase of sorption. In practice this could allow one to tune sorption/swelling and transport properties of the GO membranes directly by adjusting their internal ordering without the use of any composite materials.

## 1. Introduction

At present, the motivation to study sorption of liquids into graphite oxide (GO) materials is the possible practical use of the corresponding graphene oxide membranes for separation and purification of liquid mixtures [1,2,3,4,5,6,7,8,9,10]. Already the composite membranes, including GO, have been developed in order to tune and to increase the efficiency of separation and purification (see, e.g., [11,12]). It is known that powders of the two main types of graphite oxide, namely, of modified Hummers (HGO) and Brodie (BGO) readily swelled in polar liquids. It was once demonstrated that swelling is a key factor in determining the transport properties of the GO membranes [9]. However, the quantitative model linking swelling with the transport properties is still on demand. The swelling includes sorption of a liquid into the inter-plane space of the HGO and BGO and simultaneous increase in the inter-plane distances. The long history of syntheses of the HGO and BGO powders is described in detail in the review paper [10]. Quantitative data on sorption into graphite oxide powders was presented in [13,14,15,16], while temperature evolution of the inter-plane distances were systematically studied by A. Talyzin at al. [10,17,18,19,20]. It should be noted that sorption is considered as equilibrium value, though graphite oxides may be treated as the true equilibrium phases only conditionally. In such a case, sorption is simply stable in time reproducible value reached after sufficient contact of the adsorbent with the GO materials.

Compared to powders, relatively less information is available on the swelling of GO membranes. Some data on sorption into HGO membranes (mHGO) were presented in [15,21]. The XRD measured inter-plane distances in the mHGO and GO thin films were carefully summarized in the review paper [10]. The XRD data for the swollen BGO membranes (mBGO) may be found in [22,23,24]. The existing data make it possible to argue that membranes sorbed less and swelled less than powders. Sorption and kinetics of sorption may be influenced by significant number of factors, e.g., type of GO material, temperature, method of manufacturing of the GO membrane, chemical nature of the adsorbent, type of contact of adsorbent with the GO material, etc. The search for the governing factors determining the swelling properties of the membranes is an important task.

The present study focused on the direct comparison of sorption into the powders and corresponding membranes within each type of graphene oxide, HGO and BGO. Experimental measurements were performed under identical conditions and even simultaneously when the experimental method allowed this. The most accurate data were obtained for the difference of sorption into powders and membranes rather than for the absolute values. In this way we attempted to exclude the influence of the extraneous factors (different humidity, aging of the materials, uncontrolled chemical pre-history, different sensitivity of measurements, etc.) and to understand the real reasons making sorption into membranes relative to the powders lower. The ultimate practical goal in the future is to tune sorption properties of the pristine GO membranes in the manufacture instead of making the corresponding composites. Two effective polar adsorbents, namely acetonitrile (CH_3_CN) and water (H_2_O), were used for the comparison. Additional sorption agents chosen were 1-octanol (long-sized polar molecule), pyridine (aromatic molecule with the polar group) and benzene (non-polar liquid). It was shown that, at ambient temperature upon contact through the vapor, the membranes sorbed much less acetonitrile and pyridine compared to the powder. Direct contact with the liquid adsorbent at lower temperature resulted in equal sorption into powder and membrane. Water is equally sorbed into powders and membranes. This unusual sorption properties of H_2_O correlates with unimpeded permeation of water through the GO membranes [9]. A separate objective of the present study was to link the quantitative characteristics of the dry membrane’s alignment (orientational order parameters) with the sorption properties. These novel parameters were calculated from the electron paramagnetic resonance (EPR) spectra of the spin probes sorbed on the inner surface of the membranes [25]. Using orientational order parameters of the dry membranes as landmarks one could try to adjust sorption properties of the membrane. 

## 2. Materials and Methods

### 2.1. Synthesis of HGO and BGO Powders and Fabrication of the Membranes

HGO powders were prepared according to the modified Hummers method. The mixtures of H_2_SO_4_ and H_3_PO_4,_ with the volume ratios 9:1, 9:3 and 9:1→9:3, were applied for oxidation. In the latter case, the 9:1 mixture was used initially, and after addition of KMnO_4_ the supplemental portion of H_3_PO_4_ was added to make the volume ratio equal to 9:3. HGO powders and membranes were made from the same by-product, namely, from the aqueous dispersion of HGO obtained at the end of the modified Hummers synthetic procedure. Part of this dispersion was used to make powder, and part was used to fabricate the membrane. Water from the powder was removed by the cryochemical method. Aqueous dispersion was mixed with the liquid nitrogen and then the mixture was dried under lowered pressure. Self-standing mHGO, with a thickness of 10–50 μm, were fabricated from the same initial dispersion vacuum filtration through the alumina filters (d = 0.2 µm), or by simply drying the sonicated dispersion on a glass surface. BGO powders were synthesized according to the procedure described in [26]. The oxidation was repeated one, two or three times to make BGO(1), BGO(2) and BGO(3) samples, respectively. The mBGO were made in the same way as the mHGO. The neutral aqueous dispersions (pH = 7) of BGO were unstable. Far easier to deal with were BGO dispersions with pH = 12, which were prepared by the addition of NaOH [23]. In the present study BGO membranes were made both from the neutral and from the basic dispersions. The parameters of the dry HGO and BGO powders and membranes are given in Table 1. The thickness of the membranes prepared were at 40–50 µm. The densities, calculated from the mass and geometrical parameters of the membranes, were 1–2 g cm^−3^.

### 2.2. Instruments

DSC-30 TA from Mettler was used for measurements of sorption. The quantitative measurements relied on heating traces with the scanning rates 2 and 5 K/min. X-ray diffraction (XRD) measurements (T = 298 K) for the dry and swollen GO samples were performed with RigakuD/MAX 2500, Tokyo, Japan with CuK α radiation (λ = 1.5418 Å). XPS data were obtained using a KratosAxisUltraDLD, Kyoto, Japan instrument. EPR spectra were recorded with a Bruker EMX-500 spectrometer, Berlin, Germany. The high-sensitive resonator of Bruker ER 4119 HS was used for measurements. Thicknesses (d) of the membranes were measured with the help of micrometer with the accuracy ±5 μm.

### 2.3. Sorption Measurements 

Sorption measurements at T = 298 ± 1 K were performed by isopiestic method (IM). Equilibration of GO powders and membranes (≈5 mg), with organic liquids and water vapors, occurred within the dried desiccators and persisted until the mass of solid samples saturated with organic liquids became constant (for acetonitrile and HGO it took up to 30 days). Each saturated sample was checked by DSC for the absence of the excess of bulk organic liquid. To directly compare saturated sorption into the membrane and into the powder, the corresponding IM measurements were performed simultaneously in one desiccator. It should be emphasized that IM is designed to measure saturated sorption and is not suitable to get quantitative kinetic data. DSC sorption measurements were carried out according to the procedure described in [15]. DSC heating traces in the systems GO-sorbent were used to determine sorptions, (g/g), at melting temperature (Tm) of the sorbent. Samples were prepared by saturation/swelling of GO in the excess of liquid sorbent. This method is based on experimental observation [15,27] that liquids sorbed into the interlayer space of GO do not take part in the melting/freezing process at Tm. The mass of free liquid sorbent in direct contact with B-GO was determined from the area of the melting peak at Tm. Subtracting this mass from the total mass of the sorbent in the system provided the amount of liquid sorbed. The values of the saturated sorption obtained in this case corresponded to the melting temperatures of the liquid under study.

### 2.4. Orientational Order Parameters of Spin Probes in the Membranes

The orientational alignment of graphene oxide membranes was characterized based on the orientation distribution of the spin probes sorbed on the surface of the oxidized graphene planes inside the membranes. Stable nitroxide radicals TEMPOL and A3 were used as spin probes (Figure 1). Radicals were introduced into the membranes from the acetonitrile solutions, after that acetonitrile was removed by drying the samples on air over the course of 1–2 days. The prepared fragments of membranes with the size ~4 × 10 mm contained (1–5)·1017 spins per 1 mg of graphite oxide. Such concentrations permitted avoiding spin-exchange and dipole-dipole interactions of probing molecules. Dry samples were kept in a desiccator with P_2_O_5_. code.

To monitor the orientation distribution of radicals within the dry GO membrane, the series of spectra (15–20) were recorded at different orientations of the sample in the magnetic field (at different angles between the membrane surface normal and magnetic field lines). The experiments were performed using automatic goniometer of Bruker; the accuracy of turning was 0.5°. Simultaneous simulation of the spectral series allowed for determining the orientational order parameters of the radicals in the membrane. The simulation technique is discussed in detail in [25]. 

The order parameters P_20_, determined from the angular dependences of the EPR spectra recorded at T = 100 K (in the absence of mobility of probing molecules), are presented in Table 1. These parameters were the averaged values of the second rank Legendre functions describing the orientation of Z-axes of the g-tensors of the radicals relative to the membrane surface normal [28]. For an ideally ordered sample, P_20_ must be equal to 1; in the case of fully disordered sample, this parameter was equal to 0. It should be noted that the orientation distribution of radicals in the membranes under study were constant during several months. From the data in Table 1, it was seen that order parameters of TEMPO and A3 in the same membrane could be slightly different (see, for example, mHGO (9:3)). The reason is the stronger interaction of the aromatic molecule A3 with the surface of the oxidized graphene planes. Generally, both TEMPO and A3 adequately reflected the orientational alignment of the membranes, and parameters P_20_ could be used for comparison of ordering of the membranes prepared from various materials and fabricated using different methods. 

## 3. Results and Discussion

### 3.1. Properties of the Dry HGO and BGO Powders

Measured parameters of the dry HGO and BGO powders are presented in Table 1. As is seen from the table, minor changes in synthetic procedure for HGO did not change the XRD and XPS values. For BGO powders the monotonous increase in interplane distances (6.0 Å→6.6 Å→7.0 Å) and decrease in C/O ratio (3.3→2.8→2.5) were observed with the repeated oxidation steps. However, the changes in XRD and XPS were close to usual errors. The data obtained were close to the literature data [19]. IR spectra presented in Figure 2 qualitatively confirmed a similar composition of oxygen-containing groups in all materials under study.

### 3.2. Properties of the Dry Membranes, mHGO and mBGO

For the dry states of all powders (HGO and BGO) and corresponding membranes the inter-plain distances and C/O ratios were practically the same (see Table 1). The same peaks with the same full width at half maximum (FWHM) appeared in the XRD spectra (see, for example, Figure 3). This is consistent with the results obtained in [19,23].

The membranes prepared by vacuum filtration and simple evaporation also had the same parameters. It may be assumed that chemical composition of the material e.g., oxygen-containing groups on the surface were the same for powders and corresponding membranes. This result seems trivial for HGO material where preparation of the membranes includes only interaction with neutral water. Fabrication of mBGO included interaction with water having pH = 12. Authors [23] reported in this case a significant increase in the C/O ratio due to the removing of part of the oxygen-containing groups after interaction with NaOH; however, in [24] the same authors managed to produce mBGO with the smaller C/O = 2.7.

The rightmost column of the Table 1 presents the order parameter P_20_ of the membranes prepared. One could argue that vacuum filtration produced mHGO with rather reproducible ordering. The situation was different in the case of mBGO. Only vacuum filtration from basic dispersions allowed the attainment of the ordered BGO membranes, though with lower P20 compared to HGO. Membranes made from the neutral dispersions were completely unordered. It should be noted that the unordered membranes did not visually differ from ordered ones.

### 3.3. Sorption Properties

Table 2 gives general view of an effect of the polarity of the absorbate on sorption. The table illustrates a well-known fact that GO materials readily sorb polar liquids (molecules). The qualitative correlation between sorption and the Dimroth-Reichardt “general polarity” parameter [29] was observed. Water was easily sorbed by all powders and membranes under study. Acetonitrile was sorbed by powders HGO and BGO, membranes HGO and membrane BGO prepared from neutral dispersion. Contrary to the previously published data [13], benzene was not sorbed completely, while pyridine, aromatic molecule with the polar group, penetrated into the inter-plane space of all GO powders and HGO membranes. 1-octanol is a polar molecule with the long-chain structure. It was readily sorbed by all powders and HGO membranes. However, sorption through the vapor phase in this case was severely inhibited by the low vapor pressure of the adsorbat. Additionally, Table 2 reveals the difference between sorption properties of HGO and BGO membranes. This point will be discussed below.

Table 3 demonstrates the impact of different factors on sorption properties of GO materials. Data in the Table has proved that minor changes in the synthetic procedures for the powders, e.g., change the percentage of phosphorus acid in the synthesis of HGO or the number of oxidation steps in the case of BGO, practically did not change the values of sorption. Comparison with the literature data has demonstrated that sorption into GO powders is a reproducible and possibly the equilibrium value. In this study, sorptions into powders and membranes were directly compared (see Figure 4). The most accurate comparison was performed with simultaneous IM measurements of acetonitrile sorption into HGO powders and membranes at ambient temperature. These data are presented in Table 4 separately.

It was demonstrated that all the mHGO studied sorbed less acetonitrile than the corresponding powders (see Figure 4a). The difference in sorption for powders and membranes correlated with the difference in the corresponding inter-plane distances d001, measured by XRD for the swollen powders and membranes after IM experiments (see Table 4). The inter-plane distances obtained are in reasonable agreement with the data of ref. [22]. It is worth mentioning that in the IM sorption experiment liquid acetonitrile contacted the GO materials only through the vapor phase, while XRD measurements in [22] were performed in situ under the direct contact of GO with the adsorbed liquid. Switching from transport through the vapor to the more effective direct contact did not change the inter-plane distance in the membrane at saturation. The time required to reach the saturation was different, but the values of the inter-plane distances at saturation (i.e., swelling) were the same in both cases. It was once suggested [22] that the reasonable explanation for the reduced swelling and sorption into the membrane could be that only a fraction of the membrane was intercalated with the adsorbent, while penetration into the other part was partially restricted. In this case, one could expect that more intercalated part will produce the corresponding peak in the XRD spectra. For mHGO and acetonitrile this peak should be around 12 Å [22]. Figure 5a shows that such a peak was not found. However, the model proposed in ref. [22] is possibly realized for the sorption of pyridine into mHGO (see Figure 4b and Figure 5b). Here the reduced sorption was accompanied by the existence of the two XRD peaks at 11.4 ± 0.1 Å and 7.6 ± 0.1 Å, corresponding to the intercalated and “empty” fraction of the internal space. Lowering of the temperature increased sorption of CH_3_CN and made the saturated sorptions into the membrane and into the powder equal (see Table 3).

The sorption data presented in Table 3 and Figure 4a allows one to estimate surface areas available for acetonitrile sorption in the interplane space of mHGO. If it is assumed that increase in the interplane distance by ≈ 1 Å (see Figure 5a and ref. [20]) corresponds to the formation of a single layer of the acetonitrile molecules in the inter-plain space, one may calculate the covered surface based on the values of sorption and the value of the cross-sectional area of the CH_3_CN molecule (21.6 A^2^) [31]. The sorption area for one layer was 570 m^2^ g^−1^.

The data for mBGO are interesting and instructive. The mBGO formed from the basic solution showed some ordering, though the order parameter was smaller compared to mHGO. Sorption of acetonitrile into this membrane was practically equal to zero. It may be assumed that such membranes may completely separate acetonitrile from water. Membranes, prepared from the neutral dispersion were completely unordered (P_20_ = 0) and demonstrated some ability to sorb acetonitrile at ambient and low temperatures. One may assume that sorption properties of unordered membranes are similar to the same properties of the powders. Appearance of a non-zero order parameter pointed to the unifying of the separate particles into the common structure of the membrane, which brought about the lowering of sorption. Equal sorptions measured at a lower temperature by DSC probably point to the fact that the structure of the membrane was destroyed and the membrane was turned into the powder. It is worth mentioning that the dependence between the membrane ordering and sorption properties was observed by us earlier [25]. In this work we noted that acetonitrile introduced easier into HGO membranes with lower orientational alignment.

### 3.4. Sorption Properties of Water

Data presented in Figure 4c and Table 3 confirm unique sorption properties of water relative to GO materials. Contrary to CH_3_CN, C_5_H_5_N, CH_3_OH and C_2_H_5_OH [23], the values of saturated sorption of H_2_O into HGO powders and membranes at ambient temperature were equal. The DSC measurements of H_2_O sorption (T = 273 K) also gave identical values for the powders and the membranes. Unusual sorption properties of water correlate with its ability to exfoliate GO and to form stable liquid dispersions. Pyridine and acetonitrile lack this ability [32]. Data in Table 1 prove that the size and structure of the inter-layer space of the GO powder and the corresponding membrane were similar. The different values of sorption were caused by a different ability to penetrate into this space. It can be reasonably assumed that the rate of sorption was governed by the specific interaction of H_2_O molecule with the GO. The GO powder consisted of the individual micrometer sized particles (blocks), relatively open for penetration. The membrane was more integrated structure of the similar blocks. This hindered or completely prevented penetration, sorption and swelling for acetonitrile, pyridine, etc. but not for water. One could argue that at the moment there is no reason to consider powders and membranes to be the same material and to explain the differences in sorption into powders and membranes by certain kinetic factors. More adequate is the model of two materials with different saturated sorptions.

## 4. Conclusions

In the present study, sorption of polar substances into Hummers and Brodie graphite oxide membranes and powders were carefully compared. It was demonstrated that dry membranes and powders, though identically synthesized and showing similar XRD and XPS parameters in the dry state, had different sorption properties at ambient temperature. This difference was reproducible and did not depend upon the method of contact of polar substance with GO. However, the difference disappeared with lowering temperature and did not exist for water. Water was equally sorbed into membranes and powders at ambient and lower temperature. It was reasonable to consider sorption into the powders to be an equilibrium value. Measured sorptions into the membranes were only stable in time saturated values. They could be changed by variation of the membrane’s internal ordering. In this study the BGO membrane that did not sorb acetonitrile was prepared. Such a membrane could potentially be used for the dehumidification of acetonitrile. The EPR method, which was for the first time systematically used to characterize the set of specially prepared membranes, showed that the conventional method of preparation of the Hummers GO membranes fabricated samples with the reproducible parameter of order P_20_ ≈ 0.3–0.4. On the example of the BGO membranes, it was demonstrated that the reduction in sorption correlated with the increase in order parameters calculated from the EPR spectra. The latter result may have practical meaning. It indicates the possibility, directionally, to change sorption/swelling properties of the membranes based on the EPR measured order parameters.

## Figures and Tables

**Figure 1 membranes-13-00053-f001:**
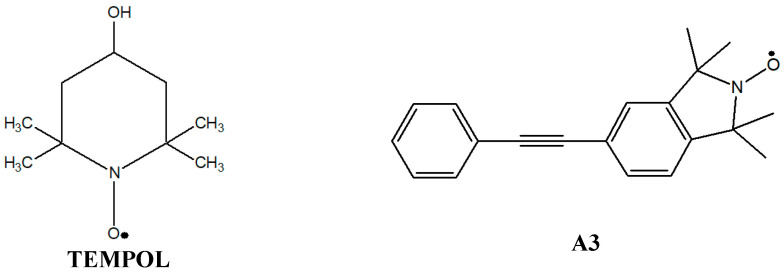
Structures of the used spin probes.

**Figure 2 membranes-13-00053-f002:**
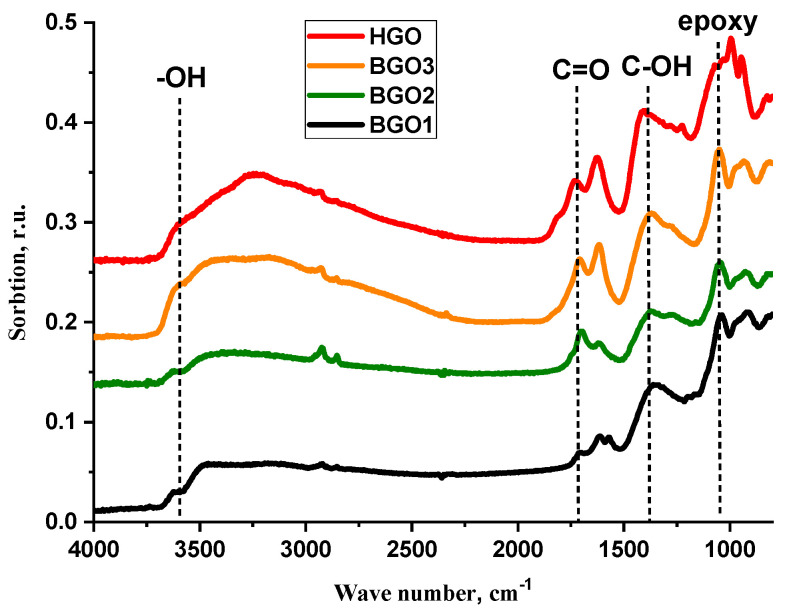
IR spectra of HGO (9:1) and BGO powders.

**Figure 3 membranes-13-00053-f003:**
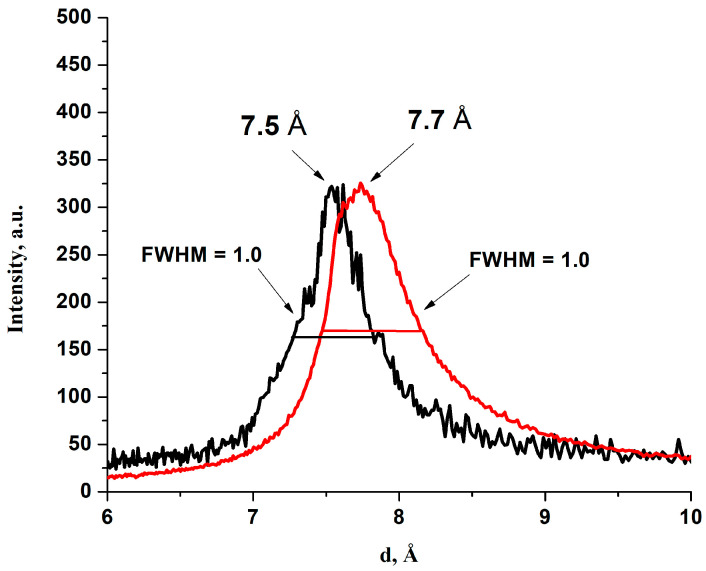
Part of the XRD spectrum of the dry HGO (9:1) powder and the corresponding membrane, prepared by evaporation.

**Figure 4 membranes-13-00053-f004:**
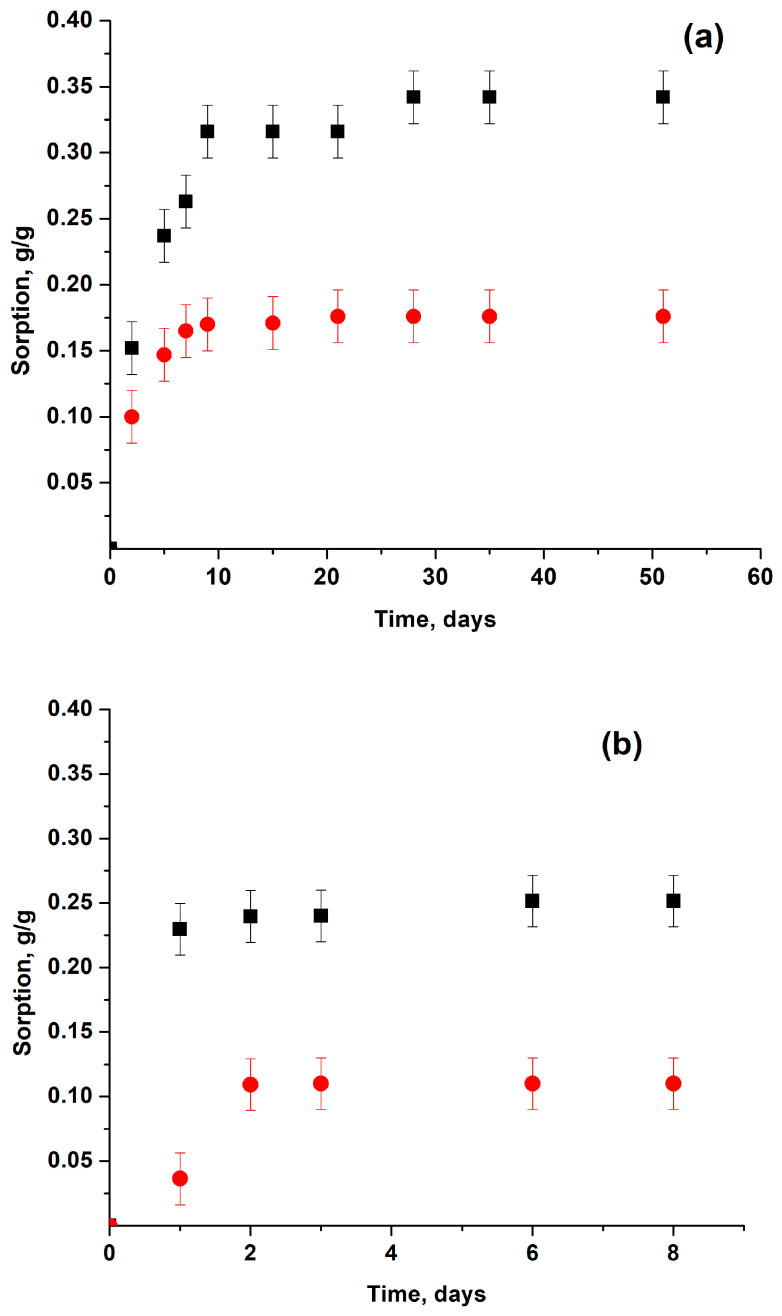
Sorption of acetonitrile (**a**), pyridine (**b**) and water (**c**) by HGO (9:3) membranes and powders. IM experiment, T = 298 K. Equilibrium sorptions correspond to the time independent parts of the curves. Black squares—powder; red circles—membrane.

**Figure 5 membranes-13-00053-f005:**
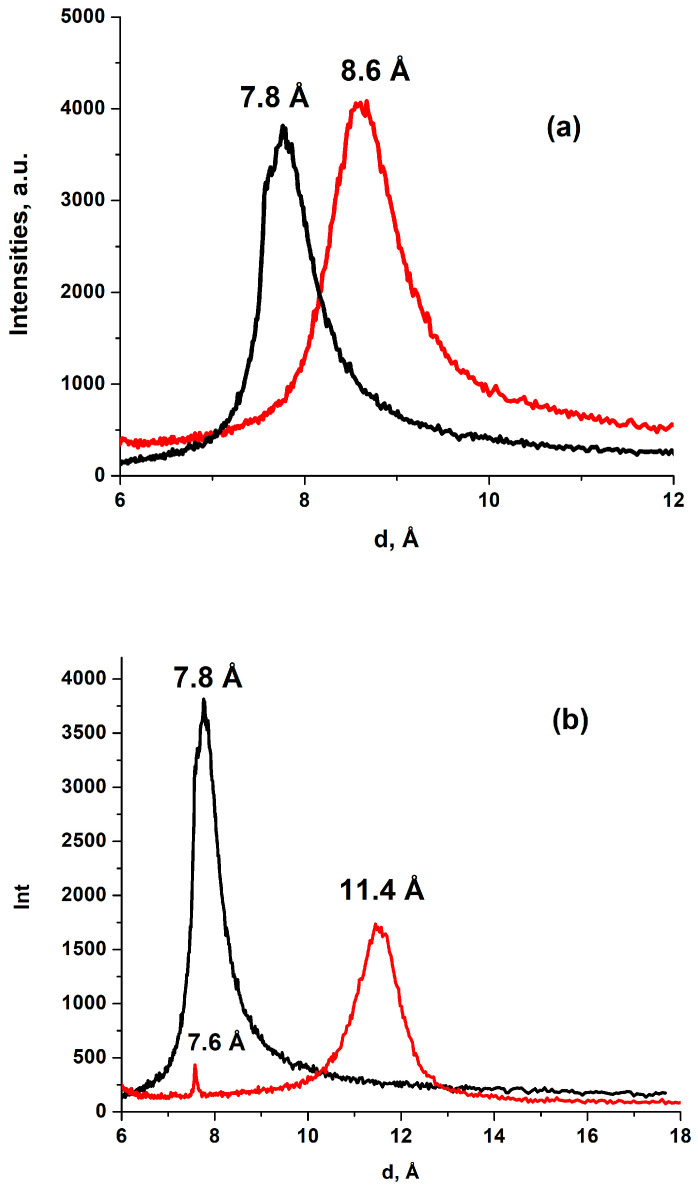
XRD of mHGO before (black line) and after (red line) sorption of acetonitrile (**a**) and pyridine (**b**).

**Table 1 membranes-13-00053-t001:** Parameters of the dry HGO and BGO powders and membranes.

GO Material	Method of the Membrane Preparation	d (001). Å ^a^ (±0.3)	C/O. ^b^ (±0.2)	P_20_ (±0.01)
HGO (9:1)		7.5	2.5	
mHGO (9:1)	Evaporation	7.7	2.5	
mHGO (9:1)	Vacuum filtration	7.8	2.5	0.33 (TEMPOL)
HGO (9:3)		7.8	2.2	
mHGO (9:3)	Vacuum filtration	7.8	2.7	0.35 (TEMPOL) 0.41 (A3)
HGO (9:1→9:3)		7.3	2.3	
mHGO (9:1→9:3)	Vacuum filtration	7.7	2.3	0.34 (TEMPOL)
HGO ^c^ [17]		7.18	2.47	
mHGO [21]	Vacuum filtration	7.7	2.49	
mHGO [9.21]	Vacuum filtration	7.57	2.81	
BGO (1)		6.0	3.3	
BGO (2)		6.6	2.8	
mBGO (2)	Vacuum filtration (pH = 12)	6.6	2.8	0.21 (A3)
mBGO (2)	Vacuum filtration (pH = 7)	6.4	2.8	≈0 (A3)
BGO (3)		7.0	2.5	
BGO [17]		6.35	2.85	
mBGO(1) [21.22]	Vacuum filtration (pH = 12)		3.84 ^d^	

^a^ Inter-plane distance, calculated from the position of d(001) peak; ^b^ XPS data; ^c^ The sample was purchased from ACS materials company and was additionally dried by the authors [19]; ^d^ This number “is consistent with possible reaction of some functional groups with the hydroxide ion and removal of this products by water” [23].

**Table 2 membranes-13-00053-t002:** Sorption (g/g) of polar and nonpolar solvents at T = 298 K.

Sorbent	^a^ ϵ	^b^ E_T_ (30)	HGO	mHGO	BGO	mBGO
CH_3_CN ^c^	38.0	45.6	0.35	0.17	0.53	^d^ 0.14; ^e^ 0
C_6_H_6_	2.2	34.3	0	0	0	0
C_5_H_5_N	1.1	40.5	0.25	0.11	0.43	0
1-octanol	9.9	48.1	0.71	0.42	1.05	0
H_2_O ^c^	78.5	63.1	0.40	0.38	0.31	0.33

^a^ Dielectric constant, ^b^ the Dimroth-Reichardt “general polarity” parameter [29], ^c^ For CH_3_CN and H_2_O sorption data are mean values. See Table 3 for details. ^d^ Sorption into the membrane, prepared from the neutral dispersion; ^e^ Sorption into the membrane, prepared from the basic dispersion.

**Table 3 membranes-13-00053-t003:** Sorption of acetonitrile and water.

Samples	Sorption of CH_3_CN		Sorption of H_2_O	
	T = 298 K	T = 219 K	T = 298 K	T = 273 K
HGO (9:1)	0.38 ± 0.02	0.47 ± 0.08	0.38 ± 0.02	0.63 ± 0.06
mHGO (9:1) Vacuum filtration	0.19 ± 0.02	0.48 ± 0.03		0.74 ± 0.02
mHGO (9:1) Evaporation	0.18 ± 0.02	0.46 ± 0.07	0.38 ± 0.02	0.61 ± 0.06
HGO (9:3)	0.34 ± 0.02		0.44 ± 0.02	
mHGO (9:3) Evaporation	0.19 ± 0.02	0.41 ± 0.03		
HGO (9:3;1)	0.33 ± 0.02		0.42± 0.02	
mHGO (9:3;1) Evaporation	0.11 ± 0.02			
HGO [15]	0.34 ± 0.02	0.48 ± 0.03		
HGO [30]			0.47± 0.03	
mHGO [21]	0.26 ± 0.02			
BGO1	0.21 ± 0.02	0.46 ± 0.07	0.29 ± 0.02	
BGO2	0.20 ± 0.02	0.54 ± 0.09	0.30 ± 0.02	0.36 ± 0.06
BGO [15]	0.25 ± 0.02	0.53 ± 0.04	0.33 ± 0.02	0.33 ± 0.02
mBGO2 Vacuum filtration (pH = 12)	<0.02	≈0	0.34 ± 0.02	0.35 ± 0.06
mBGO2 Vacuum filtration (pH = 7)	0.14 ± 0.02	0.53 ± 0.03	0.32 ± 0.02	
BGO3	0.25 ± 0.02	0.53 ± 0.05	0.31 ± 0.02	

**Table 4 membranes-13-00053-t004:** Swelling of HGO materials in CH3CN at T = 298 K.

Sample	Sorption of CH_3_CN mg/mg GO	Inter- Plane Distance, d_001, A_
HGO, powder	0.35± 0.03	12.0 ^a^
HGO, membrane	0.16± 0.03	8.8 ^a^
HGO, powder [15]	0.34± 0.01	
HGO, powder [22]		14.0 ^b^
HGO, membrane [22]		9.0 ^b^

^a^ Measured after the IM experiment, 30 days; ^b^ Measured in direct contact with liquid CH_3_ CN, ≈30 min [22].

## Data Availability

No additional data sets were generated during the study.

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
