# Peer review of "Sorption of Polar Sorbents into GO Powders and Membranes"

_membranes, 2023, doi:10.3390/membranes13010053_

Round 1

Reviewer 1 Report

1. The manuscript studies the polar substance sorption difference between HGO and BGO, but the structural difference causing the difference is not clear.  The potential influence of interplanar distance and interaction between the functional group is mentioned, but no conclusion can be drawn based on the discussion.

2. The sorption is meaningful for the filtration and separation application of the GO membrane, but what application authors are aiming for is not clear. For example, does the zero sorption of acetonitrile in BGO lead to a super high stability in the acetonitrile environment of BGO or high sorption of water lead to equal dehumidification application. Whether the specific application author is working on preferring a higher sorption rate, or a selective sorption over different polar solvent is not clear.

3. Research identified the sorption rate difference between GO membrane and powder, but structural reasons cause the difference is not well stated, not to mention the meaning of study the difference. 

4. In the first paragraph, it is important to cite the HGO and BGO you are referring to. Even though the hummus method is widely used in GO preparation, different modification to the hummus method leads to different interplanar distance of as-prepared GO. 

5. It is previous and next, before and after, pre and post.  "Previous and after" in Figure 5 caption is not proper.

Author Response

Our point-by-response to the Reviewer-1 comments are given in the attached file  

Reviewer 2 Report

The manuscript needs modifications. Please see attached file for my comments.

Author Response

Our point-by-point response to Reviewer's 2 comments are given in the attached file. 

Round 2

Reviewer 2 Report

Minor changes suggested. Please see attached file.

Author Response

Author's reply is attached
